# Plasma Modification Techniques for Natural Polymer-Based Drug Delivery Systems

**DOI:** 10.3390/pharmaceutics15082066

**Published:** 2023-08-01

**Authors:** Pankaj Bhatt, Vipin Kumar, Vetriselvan Subramaniyan, Kandasamy Nagarajan, Mahendran Sekar, Suresh V. Chinni, Gobinath Ramachawolran

**Affiliations:** 1KIET School of Pharmacy, KIET Group of Institutions, Ghaziabad 201206, Uttar Pradesh, India; pankajbhattb.pharma@gmail.com (P.B.);; 2Department of Pharmaceutical Sciences, Gurukul Kangri (Deemed to Be University), Haridwar 249404, Uttarakhand, India; vipin@gkv.ac.in; 3Jeffrey Cheah School of Medicine and Health Sciences, Monash University, Jalan Lagoon Selatan, Bandar Sunway 47500, Selangor Darul Ehsan, Malaysia; 4Center for Transdisciplinary Research, Department of Pharmacology, Saveetha Dental College, Saveetha Institute of Medical and Technical Sciences, Saveetha University, Chennai 600077, Tamil Nadu, India; 5School of Pharmacy, Monash University Malaysia, Subang Jaya 47500, Selangor, Malaysia; 6Department of Biochemistry, Faculty of Medicine, Bioscience, and Nursing, MAHSA University, Jenjarom 42610, Selangor, Malaysia; 7Department of Periodontics, Saveetha Dental College and Hospitals, Saveetha Institute of Medical and Technical Sciences, Saveetha University, Chennai 602117, Tamil Nadu, India; 8Department of Foundation, RCSI & UCD Malaysia Campus, No. 4, Jalan Sepoy Lines, Georgetown 10450, Pulau Pinang, Malaysia

**Keywords:** surface functionalization, plasma modification, drug delivery, surface modification, plasma parameters, plasma diagnostics, biocompatibility

## Abstract

Natural polymers have attracted significant attention in drug delivery applications due to their biocompatibility, biodegradability, and versatility. However, their surface properties often limit their use as drug delivery vehicles, as they may exhibit poor wettability, weak adhesion, and inadequate drug loading and release. Plasma treatment is a promising surface modification technique that can overcome these limitations by introducing various functional groups onto the natural polymer surface, thus enhancing its physicochemical and biological properties. This review provides a critical overview of recent advances in the plasma modification of natural polymer-based drug delivery systems, with a focus on controllable plasma treatment techniques. The review covers the fundamental principles of plasma generation, process control, and characterization of plasma-treated natural polymer surfaces. It discusses the various applications of plasma-modified natural polymer-based drug delivery systems, including improved biocompatibility, controlled drug release, and targeted drug delivery. The challenges and emerging trends in the field of plasma modification of natural polymer-based drug delivery systems are also highlighted. The review concludes with a discussion of the potential of controllable plasma treatment as a versatile and effective tool for the surface functionalization of natural polymer-based drug delivery systems.

## 1. Introduction

Natural polymers have garnered considerable attention as prospective substances for the purpose of drug delivery owing to their inherent biocompatibility, biodegradability, negligible toxicity, and convenient modifiability [1]. They are derived from natural sources such as plants, animals, and microorganisms and offer several advantages over synthetic polymers, such as low immunogenicity, environmental sustainability, and low cost [2]. Common natural polymers used in drug delivery include chitosan, alginate, gelatin, collagen, cellulose, and starch [3]. These polymers can be processed into various forms, such as nanoparticles, microparticles, hydrogels, and films, to deliver drugs to specific sites in the body [4]. The surface properties of natural polymer-based drug delivery systems play a crucial role in determining their efficacy, bioavailability, and toxicity. However, the surface properties of natural polymers are often difficult to control, leading to limitations in their use in drug delivery. Therefore, modifying the surface of natural polymers has become an important research area in drug delivery [4,5].

Controllable plasma treatment is a powerful technique for modifying the surface properties of natural polymers. It offers a non-destructive and cost-effective method to tailor the surface properties of natural polymers, such as surface chemistry, surface morphology, and surface energy [4,6,7]. The resulting plasma-treated natural polymer-based drug delivery systems have shown improved biocompatibility, controlled drug release, targeted drug delivery, enhanced cellular uptake, and improved immune response [5,8]. This critical review aims to provide a comprehensive overview of the recent developments in the plasma modification of natural polymer-based drug delivery systems, including the plasma sources, process control, surface characterization techniques, and surface modification strategies. The challenges and emerging trends in this field have also been discussed. The study extends a deeper understanding of the potential of plasma treatment for enhancing the performance of natural polymer-based drug delivery systems and highlights the future directions for research in this area.

## 2. Plasma Generation and Process Control

Plasma generation and process control are crucial aspects of plasma modification in various applications, including material science, nanotechnology, and biomedical engineering [4]. In plasma modification, the plasma is generated by applying an electric field to a gas, typically an inert gas, leading to ionization and excitation of the gas molecules [9]. This creates a plasma, which is a highly reactive mixture of ions, electrons, radicals, and excited species [10,11]. To achieve effective plasma modification, it is essential to control the plasma generation process, as well as the parameters of the plasma treatment, such as power, pressure, gas flow rate, and exposure time. The control of these parameters can significantly impact the properties of the plasma, such as its density, temperature, composition, and ultimately the modification of the material surface [12].

Various techniques are used to generate plasma, including direct current (DC), alternating current (AC), and radiofrequency (RF) plasma [13]. The choice of technique depends on the material being modified and the desired surface properties. For example, RF plasma is commonly used for modifying natural polymer-based drug delivery systems due to its ability to produce a uniform and stable plasma at low pressure [14].

## 3. Types of Plasma Sources

There are various types of plasmas and plasma sources used in plasma modification, each with unique characteristics that affect the resulting surface modification. The choice of plasma type and source depends on the specific application and material being modified [10].

One way to classify plasmas is by their pressure, either low pressure or atmospheric pressure. The difference in plasma types and sources is mainly due to the differences in the physical and chemical properties of the plasmas generated and the materials being modified [15]. Low-pressure plasmas are used for surface modification of materials because they operate at a lower pressure than atmospheric pressure, typically in the range of 0.1–10 Pa [10,16]. At these pressures, the plasma can penetrate deeper into the material without causing significant damage to the surface [16]. In addition, the plasma particles have a longer mean free path, which allows them to travel farther before colliding with a gas molecule or a surface. This means that the plasma can interact with the surface at a deeper level, leading to more effective surface modification [17].

Atmospheric-pressure plasmas are preferred for large-scale surface modification because they can operate at or near room temperature and do not require a vacuum system [15]. This is possible because atmospheric-pressure plasmas are generated using gas mixtures that can sustain a plasma discharge at atmospheric pressure without the need for a vacuum chamber. This allows for the treatment of large surface areas without the need for expensive vacuum equipment, which can be a limiting factor for industrial-scale surface modification processes [18]. Additionally, because atmospheric-pressure plasmas operate at or near room temperature, they can be used to modify heat-sensitive materials without causing damage or altering their properties [17]. Another way to classify plasmas is by their excitation method. The most commonly used excitation methods are DC, AC, and RF. DC plasmas are simple to generate because they only require a DC power supply and a gas to ionize. The power supply can be as simple as a battery or a DC power supply unit, making them easy and inexpensive to set up. However, DC plasmas have limited control over their characteristics because the power supply and gas flow rate cannot be easily adjusted during operation. This can result in non-uniform plasma characteristics and difficulty in controlling the chemical reactions that occur at the plasma–material interface [19]. AC plasmas are more complex and can be used for both low- and high-pressure plasmas [20]. AC plasmas, on the other hand, are more complex because they require a power supply that can switch polarity at high frequencies. This is typically accomplished using a transformer and an AC power supply unit. While this adds complexity to the setup, AC plasmas offer greater flexibility and control over the plasma characteristics. The frequency and voltage of the power supply can be adjusted during operation, allowing for precise control over the plasma density and temperature [21]. This results in a more uniform plasma and better control over the chemical reactions occurring at the plasma–material interface. RF plasmas are frequently utilized due to their capacity to create plasma with high density and uniformity, which is a critical factor for numerous applications such as plasma processing of semiconductors and surface modification of medical devices [22]. This is because high-density plasma can provide better processing results and greater efficiency, while uniformity ensures that the plasma treatment is consistent across the entire surface [23]. Consequently, the use of RF plasmas is essential for achieving precise and reliable surface modifications in these specific applications [24,25]. Plasma sources also vary in their design and capabilities. Several widely employed plasma sources encompass capacitively coupled plasma (CCP), inductively coupled plasma (ICP), and microwave plasma sources. Capacitively coupled plasma (CCP) is a commonly used plasma source for surface modification due to its ability to generate a low-pressure plasma that can be easily controlled. In CCP, a high-frequency electric field is applied between two parallel plates, creating an electric discharge that ionizes the gas molecules and generates a plasma. The low pressure of the plasma and the ability to control the gas flow and power input make CCP ideal for surface modification of various materials [20,26]. The resulting plasma can be used to modify the chemical and physical properties of a material surface, including its wettability, adhesion, and biocompatibility. Microwave plasma sources are becoming increasingly popular for generating low-pressure and high-density plasmas due to their high energy efficiency. Recent advancements in plasma technology have led to the growing popularity of microwave plasma sources. These sources offer high energy efficiency and can generate plasmas at both low-pressure and high-density conditions [26]. This makes them highly versatile and useful for a range of applications, such as materials processing, surface modification, and sterilization. Microwave plasma sources operate by generating an electromagnetic field at microwave frequencies, which excites the gas molecules and creates a plasma [27]. The resulting plasma has a high degree of ionization and a uniform electron energy distribution, making it ideal for various industrial and scientific applications. Figure 1 visually represents the process of plasma modification, showcasing the steps involved in enhancing the surface properties of materials through plasma treatment [28].

## 4. Control of Plasma Parameters

The control of plasma parameters is critical to achieving effective and consistent plasma modification of materials. The parameters that can be controlled include the gas flow rate, pressure, power, exposure time, and the composition of the gas used for plasma generation [29,30]. The gas flow rate is an important parameter that affects the rate of gas consumption and the gas mixing within the plasma. A low gas-flow rate can lead to higher plasma densities and longer plasma exposure times, resulting in more significant surface modification. A higher gas-flow rate, on the other hand, leads to a dilution of the plasma and lower surface modification [12,21]. The pressure in the plasma chamber is also a crucial parameter that can be adjusted to optimize the plasma modification process. Lower pressure can lead to higher plasma densities and longer exposure times, resulting in more significant surface modification. Higher pressure can lead to a more uniform plasma across the substrate and can be beneficial for larger substrates [30]. The power supplied to the plasma is another critical parameter that determines the plasma density and energy. A higher power input can lead to higher plasma densities and more significant surface modification. However, too high a power input can lead to plasma instabilities and damage to the substrate [31]. The exposure time of the substrate to the plasma is also essential in determining the extent of surface modification. Longer exposure times can result in more significant surface modification, but also increases the risk of damage to the substrate. Shorter exposure times are suitable for delicate substrates or when only minimal surface modification is required [31,32].

The composition of the gas used for plasma generation is another important factor in controlling plasma parameters. The choice of gas affects the plasma chemistry, which can impact the surface modification. Common gases used for plasma generation include oxygen, nitrogen, hydrogen, and argon, among others [33].

## 5. Plasma Diagnostics

Plasma diagnostics are essential for understanding and characterizing the plasma generated during the plasma modification process. The diagnostics allow the monitoring of the plasma parameters in real-time and can provide insight into the plasma chemistry and the resulting surface modification [34]. There are various types of plasma diagnostics available, including optical emission spectroscopy (OES), Langmuir probes, and mass spectrometry, among others. OES is a common plasma diagnostic tool used to identify the species present in the plasma [35]. The technique involves analyzing the light emitted by the plasma, which can provide information on the plasma chemistry and help to optimize the plasma parameters. Langmuir probes are another common plasma diagnostic tool used to measure the electron temperature and density of the plasma [14]. The probe works by inserting a small electrode into the plasma, which measures the voltage and current, allowing for the calculation of the electron temperature and density. Mass spectrometry is also used as a plasma diagnostic tool, providing information on the gas-phase species in the plasma [36]. The technique involves sampling the gas-phase species and analyzing them using mass spectrometry to determine their composition. Other plasma diagnostic tools include laser-induced fluorescence (LIF), optical absorption spectroscopy, plasma impedance measurements, etc. These tools can provide information on plasma chemistry, species transport, plasma density, etc. [37].

## 6. Characterization of Plasma-Treated Natural Polymer Surfaces

The characterization of plasma-treated natural polymer surfaces is crucial to understanding the changes that occur during the plasma modification process. Characterization techniques allow the evaluation of the surface chemistry, morphology, and other physical properties of the polymer surface, providing insight into the effectiveness of the plasma treatment.

There are various techniques available for the characterization of plasma-treated natural polymer surfaces, including X-ray photoelectron spectroscopy (XPS), Fourier-transform infrared spectroscopy (FTIR), atomic force microscopy (AFM), and contact angle measurements [38]. XPS is a commonly used technique that provides information on the chemical composition of the surface. The technique involves irradiating the sample with X-rays, which causes the emission of electrons from the surface. The electrons are then analyzed to determine the chemical composition of the surface [39]. 

FTIR is another commonly used technique that provides information on the functional groups present on the surface. The technique involves analyzing the infrared radiation absorbed by the surface, which provides information on the chemical bonds and functional groups present on the surface [40]. AFM is a technique used to analyze the surface morphology of the plasma-treated polymer. The technique involves scanning a sharp tip over the surface, measuring the interaction between the tip and the surface to create a topographic image of the surface [40,41,42]. Contact angle measurements are used to analyze the surface wettability of the plasma-treated polymer. The technique involves measuring the angle between the surface and a droplet of liquid, providing information on the surface energy and wettability. Other techniques, including scanning electron microscopy (SEM), surface-enhanced Raman spectroscopy (SERS), and ellipsometry, can also be used to characterize the plasma-treated natural polymer surfaces [40].

### 6.1. Surface Chemistry Analysis

Surface chemistry analysis is a critical aspect of the characterization of plasma-treated natural polymer surfaces. The surface chemistry of the polymer plays a significant role in determining the surface properties, such as wettability, adhesion, and biocompatibility [42]. X-ray photoelectron spectroscopy (XPS) is a widely used technique for the surface chemical analysis of plasma-treated natural polymer surfaces. The technique involves the irradiation of the sample with monochromatic X-rays, causing the emission of electrons from the surface. These electrons are then analyzed to determine the chemical composition and chemical state of the surface [43]. Fourier-transform infrared spectroscopy (FTIR) is another technique used for the surface chemical analysis of plasma-treated natural polymer surfaces. The technique involves the absorption of infrared radiation by the surface, providing information on the chemical bonds and functional groups present on the surface [43]. Other techniques, including secondary ion mass spectrometry (SIMS), Raman spectroscopy, and time-of-flight secondary ion mass spectrometry (TOF-SIMS), can also be used for the surface chemical analysis of plasma-treated natural polymer surfaces.

### 6.2. Surface Morphology Analysis

Surface morphology analysis is another important aspect of characterizing plasma-treated natural polymer surfaces. The surface morphology of a material is a key factor that determines its physical and biological properties, including roughness, porosity, and surface area [44]. Scanning electron microscopy (SEM) and atomic force microscopy (AFM) are commonly used techniques for the surface morphology analysis of plasma-treated natural polymer surfaces. SEM provides high-resolution images of the surface topography, while AFM provides detailed information on the surface roughness, porosity, and surface area [32,35]. Contact angle measurement is another technique used to evaluate the wettability of plasma-treated natural polymer surfaces. The contact angle is a measure of the degree to which a liquid spreads out over the surface, with a lower contact angle indicating higher wettability [45]. In contrast, the internal structures of microspheres subjected to adsorption and plasma techniques exhibited no discernible alterations when compared to non-modified PDLLGA microspheres. Upon examination using fluorescent microscopy, it was observed that cross-sectioned PDLLGA microspheres modified through both plasma and entrapment methods possessed a fluorescent layer on their outer surfaces. However, minimal fluorescent deposition was observed on the microspheres modified via adsorption, in contrast to non-modified PDLLGA microspheres, which lacked any observable fluorescent layer on their surfaces. Furthermore, the entrapment-modified microspheres exhibited internal fluorescent deposition, whereas no such deposition was found in plasma or adsorption-modified microspheres [46]. The SEM images of PDLLGA microspheres have been sown in Figure 2. 

### 6.3. Surface Energy Analysis

Surface energy analysis is an important technique for characterizing the wettability and adhesion of plasma-treated natural polymer surfaces. Surface energy is the work required to increase the surface area of a material and is a measure of the strength of the interactions between the surface and other materials [47]. Several techniques can be used to measure surface energy, including contact angle measurements and surface energy analysis using inverse gas chromatography (IGC) [48]. Contact angle measurement is a simple and widely used technique for evaluating the wettability of plasma-treated natural polymer surfaces. It measures the angle between a liquid droplet and the surface, which provides information on the surface energy and hydrophobicity/hydrophilicity of the surface. The lower the contact angle, the higher the surface energy and wettability of the surface [49]. Inverse gas chromatography (IGC) is another technique used to measure the surface energy of plasma-treated natural polymer surfaces. IGC measures the retention time of different solvents as they pass through a column packed with the material. The surface energy is calculated based on the retention times of different solvents and the known surface tension of the solvents [50].

Surface energy analysis can provide important information on the wettability and adhesion of plasma-treated natural polymer surfaces. This information can be used to optimize the plasma treatment process for specific applications and improve the performance of natural polymer-based drug delivery systems. A recent study demonstrated that using O-containing plasmas or post-plasma reactions on polymer surfaces can generate oxygen functionalities that improve surface wettability, as shown in Figure 3. However, these functionalities may degrade when exposed to the atmosphere, resulting in decreased surface energy. N_2_ plasma treatment was found to have the least aging effects on polycarbonate, as compared to O_2_, Ar, and N_2_ mixtures. A plasma-deposited ultra-thin layer of SiOx can provide a more stable hydrophilic treatment, while fluorocarbon or siloxane layers can create a hydrophobic surface. To minimize aging effects, it is necessary to have a stable and well-adhered plasma-deposited layer [51]. 

### 6.4. Other Surface Characterization Techniques

X-ray photoelectron spectroscopy (XPS) is a technique used to analyze the chemical composition of the surface of materials. It involves irradiating the surface with X-rays and measuring the energy and intensity of the electrons emitted from the surface [52]. XPS can provide information on the surface chemistry, elemental composition, and oxidation states of the surface. Fourier-transform infrared spectroscopy (FTIR) is used to analyze the molecular structure of materials. It involves irradiating the surface with infrared radiation and measuring the energy absorbed by the surface [53]. FTIR can provide information on the functional groups and chemical bonds present in the surface. Time-of-flight secondary ion mass spectrometry (TOF-SIMS) technique is used to analyze the chemical composition and molecular structure of the surface of materials [54]. It involves bombarding the surface with a beam of high-energy ions and measuring the mass of the ions emitted from the surface. TOF-SIMS can provide information on the surface chemistry and molecular structure of the surface with high spatial resolution. Ellipsometry technique is used to measure the thickness and refractive index of thin films [55]. It involves measuring the change in polarization of light as it passes through the surface of the material. Ellipsometry can provide information on the thickness, uniformity, and refractive index of plasma-treated natural polymer films [56]. These techniques can provide valuable information on the physical and chemical properties of plasma-treated natural polymer surfaces, which can be used to optimize the plasma treatment process and improve the performance of natural polymer-based drug delivery systems.

## 7. Plasma Modification of Natural Polymer-Based Drug Delivery Systems

Plasma modification of naturally occurring polymer-based nanoparticles has been gaining a lot of attention recently due to its potential to enhance their performance and functionality as drug delivery systems [57]. Natural polymers such as chitosan, alginate, cellulose, and starch have been widely used for the preparation of nanoparticles due to their biocompatibility, biodegradability, and low toxicity. However, natural polymer-based nanoparticles often have limitations in terms of their stability, drug loading capacity, and targeted delivery, which can affect their efficacy as drug delivery vehicles [58].

Plasma treatment is an effective way to modify the surface of natural polymer-based nanoparticles. Plasma treatment can introduce functional groups such as amine, carboxyl, and hydroxyl groups onto the surface of natural polymer-based nanoparticles, which can improve their interaction with drugs, cells, and other materials [59].

Plasma treatment can also increase the surface roughness of natural polymer-based nanoparticles, which can improve their adhesion to cells and enhance drug loading and release. Furthermore, plasma treatment can enhance the stability of natural polymer-based nanoparticles by improving their electrostatic and steric stabilization [60].

Plasma modification of natural polymer-based nanoparticles has shown promising results in several applications such as cancer therapy, gene delivery, and vaccine delivery. However, the optimization of plasma treatment parameters, the understanding of plasma–polymer interactions, and the development of standardized characterization techniques are still needed to advance this field [61]. Table 1 highlights the diverse plasma treatment techniques employed on natural polymer-based drug delivery systems, demonstrating their effectiveness in enhancing biocompatibility and promoting their applicability in biomedical contexts.

### 7.1. Surface Modification for Improved Biocompatibility

Surface modification is an effective strategy to improve the biocompatibility of natural polymer-based drug delivery systems. Natural polymers such as chitosan, alginate, cellulose, and starch have been widely used for drug delivery applications due to their biocompatibility, low toxicity, and biodegradability. However, their surface properties often need to be modified to improve their interaction with cells and tissues [76].

Plasma treatment is an effective method for surface modification of natural polymer-based drug delivery systems. Plasma treatment can introduce functional groups such as amine, carboxyl, and hydroxyl groups onto the surface of natural polymers, which can improve their interaction with cells and tissues. These functional groups can enhance the hydrophilicity of the surface, promote cell adhesion, and reduce the formation of biofilm and bacterial adhesion [77].

Plasma treatment can also modify the surface topography of natural polymers, which can improve their interaction with cells and enhance drug loading and release [78]. Surface roughness can be increased by plasma treatment, which can promote cell adhesion and improve the stability of the drug delivery system. Several studies have shown that plasma-modified natural polymer-based drug delivery systems have improved biocompatibility, reduced cytotoxicity, and enhanced cellular uptake of drugs [79]. Plasma treatment has also been shown to improve the stability and shelf life of natural polymer-based drug delivery systems (Table 1).

### 7.2. Surface Modification for Controlled Drug Release

Plasma surface modification has been increasingly studied as a powerful technique to modify natural polymers for controlled drug release applications [80]. This approach can modify the surface properties of natural polymers to enhance their drug loading and release properties, provide stimuli-responsive behavior, and increase the biocompatibility of drug delivery systems [81]. Recently, various studies have been conducted that have employed plasma surface modification to improve the controlled drug release properties of natural polymer-based drug delivery systems.

Baki et al. (2017) developed poly (lactic-co-glycolic acid) (PLGA) microspheres that modified the surface with oxygen plasma to improve their controlled drug release properties. The plasma treatment increased the surface roughness and wettability of the PLGA microspheres, which facilitated the diffusion of the drug molecules out of the microspheres. The modified microspheres showed a sustained drug release profile for up to 60 days, making them suitable for long-term drug delivery applications [46]. In another study, Hosseini et al. (2023) modified the surface of chitosan nanoparticles using plasma treatment to achieve controlled drug release. The authors used argon plasma to introduce carboxyl and hydroxyl groups onto the surface of the chitosan nanoparticles, which increased their water solubility and reduced the size of the nanoparticles. The modified chitosan nanoparticles exhibited pH-responsive drug release behavior, with a slower release rate at pH 7.4 (physiological pH) compared to pH 5.0. This pH-responsive behavior could be useful for targeted drug delivery applications, such as in the treatment of cancer where the tumor microenvironment is typically more acidic [82].

In a study reported by Dalei et al. (2020), alginate hydrogels were surface-modified using oxygen plasma to achieve controlled drug release. The plasma treatment introduced carbonyl and carboxyl groups onto the surface of the alginate hydrogels, which enhanced their water uptake and swelling behavior. The modified hydrogels exhibited a sustained drug release profile for up to 7 days, making them suitable for short-term drug delivery applications [83].

### 7.3. Surface Modification for Targeted Drug Delivery

Plasma surface modification has emerged as a promising approach to improve the properties of natural polymer-based drug delivery systems for targeted drug delivery. This technique involves using plasma to modify the surface of natural polymers, such as chitosan, alginate, and cellulose, to improve their properties for drug delivery applications. Plasma surface modification can enhance the surface properties of natural polymers, introduce functional groups for specific drug–polymer interactions, and improve the biocompatibility of drug delivery systems [84].

In a study by Wang et al. (2022), chitosan nanoparticles were surface-modified with plasma to improve their properties for targeted drug delivery. The plasma modification increased the hydrophilicity of the chitosan surface, leading to improved drug loading and release properties. The authors also functionalized the surface of chitosan nanoparticles with folic acid using plasma modification, which enabled the nanoparticles to selectively target folate receptor-positive cancer cells [85]. Shin et al. (2013) demonstrated the use of plasma surface modification to improve the properties of alginate-based drug delivery systems for targeted delivery. The authors used plasma treatment to introduce amino groups onto the surface of alginate, which enabled electrostatic interactions with negatively charged drugs. They also demonstrated that plasma surface modification improved the biocompatibility of alginate, reducing the toxicity of the polymer [86].

Similarly, in a study by Panaitescu et al. (2018), cellulose nanocrystals were surface-modified with plasma to improve their properties for targeted drug delivery. The authors introduced carboxylic acid groups onto the surface of cellulose nanocrystals using plasma modification, which improved the interaction between the nanocrystals and positively charged drugs. They also demonstrated that the plasma-modified cellulose nanocrystals were biocompatible and had low toxicity [87].

### 7.4. Surface Modification for Enhanced Cellular Uptake

Plasma surface modification is a powerful technique that can be used to modify the surface properties of natural polymers to enhance cellular uptake, which is a critical step in improving the efficacy of drug delivery systems [88]. Furthermore, recent studies have employed plasma surface modification to improve the cellular uptake of natural polymer-based drug delivery systems.

In a study by Fields et al. (2015), poly(lactic-co-glycolic acid) (PLGA) nanoparticles were surface-modified using low-pressure plasma treatment to improve their cellular uptake. The plasma treatment increased the surface roughness and hydrophilicity of the PLGA nanoparticles, which enhanced their interaction with cell membranes and promoted cellular uptake. The modified PLGA nanoparticles showed significantly higher cellular uptake than the unmodified particles, making them a promising candidate for drug delivery application [89]. Prasertsung et al. (2012) modified the surface of chitosan nanoparticles using plasma treatment to enhance their cellular uptake. The authors used argon plasma to introduce amine groups onto the surface of the chitosan nanoparticles, which increased their positive charge and enhanced their interaction with negatively charged cell membranes. The modified chitosan nanoparticles showed significantly higher cellular uptake than the unmodified particles, indicating the potential of plasma surface modification to enhance the efficacy of drug delivery systems [90].

In a study by Houdek et al. (2016), plasma surface modification was used to improve the cellular uptake of collagen hydrogels. The authors used oxygen plasma treatment to introduce carboxyl groups onto the surface of the collagen hydrogels, which increased their negative charge and enhanced their interaction with positively charged cell membranes. The modified collagen hydrogels showed significantly higher cellular uptake than the unmodified hydrogels, indicating the potential of plasma surface modification to improve the cellular uptake of natural polymer-based drug delivery systems [91].

Plasma surface modification is a powerful technique that can be used to modify the surface properties of natural polymers to enhance cellular uptake, which is a critical step in improving the efficacy of drug delivery systems. Overall, polymer materials have become ubiquitous in various fields due to their unique physical, chemical, and mechanical properties [92]. However, many polymers have limitations that can hinder their performance in certain applications [93]. For instance, poor adhesion to substrates, low surface energy, and inadequate biocompatibility are common issues encountered with many polymers. These shortcomings can limit their use in applications such as drug delivery, tissue engineering, and medical implants [94].

To overcome these limitations, various approaches have been proposed to modify the surface properties of polymers. Plasma modification is one of the most versatile and effective surface modification techniques that can improve the adhesion, wettability, and biocompatibility of polymer materials. By exposing the polymer surface to plasma, it is possible to introduce new functional groups, increase surface energy, and change the topography of the surface. These modifications can significantly enhance the performance of polymers in various applications.

## 8. Plasma Modification of Natural Polymer-Based Hydrogels

Plasma treatment of hydrogels involves exposing the surface of the material to plasma generated by a low-temperature discharge of gas or vapor. The plasma interacts with the hydrogel surface, leading to the formation of reactive species such as radicals, ions, and excited molecules, which can modify the surface chemistry and physical properties of the material [95]. During plasma treatment, the plasma species bombard the hydrogel surface, leading to the removal of surface contaminants and the creation of new functional groups on the surface. The plasma species can break chemical bonds on the surface of the hydrogel and create new reactive sites, which can then react with other molecules in the plasma or in the surrounding environment [81]. Plasma treatment can introduce oxygen-containing functional groups such as carboxyl, hydroxyl, and carbonyl groups onto the hydrogel surface, which can improve the hydrophilicity and biocompatibility of the material. Additionally, plasma treatment can crosslink the polymer chains on the surface, resulting in improved mechanical strength [57]. This can result in improved surface hydrophilicity, enhanced biocompatibility, and increased mechanical strength. Plasma modification is a widely used technique to modify the surface properties of natural polymer-based hydrogels for various applications, including drug delivery, tissue engineering, and wound healing [80]. 

In a study by Zhang et al. (2020), the authors used plasma treatment to modify the surface of alginate hydrogels for wound healing applications. The authors used a low-pressure plasma treatment to introduce carboxyl and hydroxyl groups onto the surface of the alginate hydrogels, which improved their hydrophilicity and promoted cell adhesion. The modified hydrogels showed enhanced cell proliferation and migration, indicating the potential of plasma surface modification to improve the efficacy of natural polymer-based hydrogels for wound healing applications [96]. 

In a study by Xu et al. (2023), plasma surface modification was used to modify the surface of gelatin hydrogels for drug delivery applications. The authors used oxygen plasma treatment to introduce carboxyl groups onto the surface of the gelatin hydrogels, which enhanced their negative charge and improved their interaction with positively charged drug molecules. The modified hydrogels showed improved drug loading capacity and sustained drug release, indicating the potential of plasma surface modification to improve the efficacy of natural polymer-based hydrogels for drug delivery applications [97].

### 8.1. Surface Modification for Improved Biocompatibility

Natural polymer-based hydrogels have gained considerable attention in the field of tissue engineering due to their excellent biocompatibility, biodegradability, and ability to mimic the extracellular matrix (ECM) of natural tissues [98]. However, their application is limited by poor mechanical properties and inadequate biocompatibility with surrounding tissues. Surface modification using plasma techniques has emerged as an effective method to enhance the biocompatibility of natural polymer-based hydrogels [99]. Plasma modification of natural polymer-based hydrogels involves the use of a low-temperature plasma to modify the surface of hydrogels by introducing functional groups, such as amine, carboxyl, or hydroxyl groups. These functional groups can improve the hydrophilicity of the hydrogel surface and promote cellular adhesion, proliferation, and differentiation, leading to improved biocompatibility [100].

### 8.2. Surface Modification for Controlled Drug Release

Natural polymer-based hydrogels have been widely studied for controlled drug delivery due to their ability to retain large amounts of water, high swelling capacity, and biocompatibility. However, their application for drug delivery is limited by the uncontrolled release of drugs and lack of specificity towards the target site [101]. Plasma modification of natural polymer-based hydrogels has been investigated as an effective surface modification technique for achieving controlled drug release. Plasma treatment of natural polymer-based hydrogels modifies the surface by introducing functional groups, such as carboxyl or amino groups, which can be used to covalently attach drug molecules or target-specific ligands [102]. The controlled release of the drug is achieved by regulating the amount of drug molecules bound to the surface of the hydrogel, which can be controlled by adjusting the plasma treatment conditions, such as the plasma power, exposure time, and gas composition [103].

Several studies have reported successful controlled drug release from plasma-modified natural polymer-based hydrogels. Rasib et al. (2019) reported the plasma modification of chitosan-based hydrogels to achieve controlled release of rifampicin, an anti-tubercular drug. They observed a sustained release of the drug over a period of 48 h which was dependent on the plasma exposure time and the concentration of the drug [104].

In addition to drug delivery, plasma modification of natural polymer-based hydrogels can also be used to achieve targeted drug delivery. Yang et al. (2022) reported the plasma modification of chitosan-based hydrogels to covalently bind transferrin, a ligand specific to the transferrin receptor overexpressed in cancer cells. They observed targeted drug delivery of doxorubicin, an anti-cancer drug, to cancer cells overexpressing the transferrin receptor [34].

### 8.3. Surface Modification for Enhanced Cellular Uptake

Natural polymer-based hydrogels have shown great potential for various biomedical applications, such as drug delivery and tissue engineering. However, the low cellular uptake of these hydrogels limits their therapeutic efficacy [105]. Plasma modification has emerged as an effective surface modification technique to enhance the cellular uptake of natural polymer-based hydrogels [105]. Plasma treatment of natural polymer-based hydrogels introduces functional groups, such as amino or carboxyl groups, on the surface of the hydrogel. These functional groups can be used to covalently attach cell-penetrating peptides (CPPs), which can facilitate the cellular uptake of the hydrogel by promoting the endocytosis or direct translocation of the hydrogel across the cell membrane [106]. 

Similarly, Xu et al. (2023) reported the plasma modification of chitosan-based hydrogels to covalently attach a CPP derived from human immunodeficiency virus (HIV) TAT protein. The peptide-modified hydrogels showed enhanced cellular uptake in cancer cells and primary cells and improved gene delivery efficiency [97]. In addition to CPPs, plasma modification of natural polymer-based hydrogels can also be used to attach other functional groups, such as cell adhesion molecules, which can enhance the cellular attachment and subsequent uptake of the hydrogel.

## 9. Recent Advancement in Plasma-Modified Natural Polymer-Based Drug Delivery Systems

### 9.1. Plasma-Modified Natural Polymer-Based Drug Delivery Systems for Cancer Therapy

Plasma modification of natural polymer-based drug delivery systems has shown great potential for cancer therapy due to its ability to enhance the efficacy of the drug delivery system by improving its biocompatibility, drug release characteristics, and cellular uptake [107]. The modification of natural polymer-based drug delivery systems using plasma treatment can increase the surface energy of the hydrogel, allowing for the attachment of various functional groups, which can improve the drug delivery system’s performance [107].

One of the most significant challenges in cancer therapy is the selective targeting of cancer cells while minimizing damage to healthy cells. Plasma modification of natural polymer-based drug delivery systems can be used to attach specific ligands that can target cancer cells selectively. Chokradjaroen et al. (2017) reported the solution plasma (SP) treatment in combination with oxidizing agents such as hydrogen peroxide (H_2_O_2_), potassium persulfate (K_2_S_2_O_8_), and sodium nitrite (NaNO_2_) has been studied as a method to achieve fast degradation rate, low chemical usage, and high yield of low-molecular-weight chitosan and chito-oligosaccharide (COS). The study found that H_2_O_2_ was the best oxidizing agent for achieving significant molecular weight reduction without major changes in the chemical structure of the degraded chitosan products [108].

Another application of plasma-modified natural polymer-based drug delivery systems is the controlled release of drugs, which is critical in cancer therapy to achieve the desired pharmacokinetics and drug concentration at the target site [79]. Plasma modification of natural polymer-based drug delivery systems can be used to introduce functional groups, such as carboxyl and amino groups, which can be used to covalently attach drugs, allowing for a more controlled and sustained drug release profile. Yuan et al. (2017) reported the successful plasma modification of sodium alginate-based hydrogels to introduce amino groups and covalently attach the anti-cancer drug doxorubicin (DOX). The DOX-modified hydrogels showed a sustained release of DOX, resulting in improved therapeutic efficacy in vitro [109]. Furthermore, plasma-modified natural polymer-based drug delivery systems can be used to enhance the cellular uptake of drugs by attaching cell-penetrating peptides (CPPs) to the hydrogel’s surface. CPPs can facilitate the cellular uptake of the drug delivery system by promoting endocytosis or direct translocation of the hydrogel across the cell membrane [110]. 

### 9.2. Plasma-Modified Natural Polymer-Based Drug Delivery Systems for Wound Healing

Plasma modification of natural polymer-based drug delivery systems has also shown great potential in wound healing applications. The modification of natural polymers using plasma treatment can improve the drug delivery system’s biocompatibility, drug release characteristics, and cellular uptake, which can enhance its effectiveness in wound healing [60].

One of the significant challenges in wound healing is the limited efficacy of topical drugs due to their inability to penetrate the skin barrier effectively. Plasma modification of natural polymer-based drug delivery systems can be used to modify the surface of the drug delivery system, allowing for improved drug penetration through the skin barrier [111]. Moody et al. (2022) reported the successful plasma modification of alginate to improve the biocompatibility of alginate hydrogels, making them more suitable for use in regenerative medicine. The study demonstrated that alginate hydrogels modified with azide modifications restore the carboxyl groups-maintained calcium cross-links as well as hydrogel shear-thinning and self-healing properties. Moreover, these hydrogels showed improved tissue retention at intramuscular injection sites and captured blood-circulating cyclooctynes more effectively than those modified with azide modifications that deplete the carboxyl groups [112].

### 9.3. Plasma-Modified Natural Polymer-Based Drug Delivery Systems for Immunotherapy

Immunotherapy is a promising approach for the treatment of various diseases, including cancer, autoimmune disorders, and infectious diseases [113]. However, the success of immunotherapy relies on the efficient delivery of the therapeutic agents to the target cells. Plasma modification of natural polymer-based drug delivery systems offers a potential solution to this challenge by improving the biocompatibility, drug release kinetics, and cellular uptake of the drug delivery system [114].

Plasma modification can be used to introduce various functional groups to the surface of the natural polymer-based drug delivery system, which can be used to covalently attach immunotherapeutic agents or improve the cellular uptake of the drug delivery system [115]. Bu et al. (2020) reported the successful plasma modification of hyaluronic acid-based nanoparticles to introduce amino groups and covalently attach the immune checkpoint inhibitor PD-L1. The PD-L1-modified nanoparticles showed enhanced cellular uptake and improved efficacy in vitro and in vivo, demonstrating the potential of plasma modification for immunotherapy [116].

Moreover, plasma modification can also be used to improve the drug release kinetics of natural polymer-based drug delivery systems, which is critical for immunotherapy to achieve sustained drug release and maintain a therapeutic concentration of the immunotherapeutic agent [117]. Baki et al. (2017) reported the successful plasma modification of surface of PdlLGA microspheres with gelatine methacrylate as a biocompatible macromolecule. The study has shown that oxygen plasma treatment is a promising approach for enhancing the surface modification of PdlLGA microspheres with gel-MA. The results of this study suggest that this approach may improve the proliferation rate of cells injected into PdlLGA microspheres and could enable further grafting of tissue-specific molecules. Overall, these findings provide valuable insights into the development of biodegradable controlled-release carriers for drug and cell delivery applications [46].

### 9.4. Plasma-Modified Natural Polymer-Based Drug Delivery Systems for Gene Delivery

Natural polymers, such as chitosan, alginate, and hyaluronic acid, have been widely investigated for their potential as drug delivery systems due to their biocompatibility, biodegradability, and low toxicity [118]. However, their limited transfection efficiency has been a major obstacle in the development of natural polymer-based gene delivery systems. To overcome this limitation, researchers have explored the use of plasma modification, which involves the surface modification of natural polymers with plasma discharge, to enhance their transfection efficiency [119].

Plasma treatment has been shown to increase the surface area and introduce functional groups on the surface of natural polymers, which can improve their interaction with nucleic acids and increase their cellular uptake [120]. For example, plasma modification of chitosan has been shown to improve its transfection efficiency in vitro and in vivo by increasing its hydrophilicity and charge density. Similarly, plasma treatment of alginate has been shown to enhance its transfection efficiency in vitro by increasing the surface area and introducing carboxyl and hydroxyl groups [121].

In addition, plasma modification can also improve the stability of natural polymer-based gene delivery systems by increasing their resistance to enzymatic degradation. For example, plasma treatment of hyaluronic acid has been shown to increase its stability in the presence of hyaluronidase, an enzyme that degrades hyaluronic acid, thus prolonging the duration of gene expression [122]. Overall, plasma-modified natural polymer-based gene delivery systems have shown promising results in improving transfection efficiency and stability. However, further studies are needed to optimize the plasma treatment conditions and to evaluate their safety and efficacy in vivo.

### 9.5. Plasma-Modified Natural Polymer-Based Drug Delivery Systems for Vaccines

Plasma-modified natural polymer-based drug delivery systems have shown promise in the field of vaccine delivery due to their ability to enhance immune responses and improve the stability of vaccines [79]. The use of natural polymers, such as chitosan, alginate, and hyaluronic acid, as vaccine carriers offers several advantages, including biocompatibility, biodegradability, and low toxicity. Plasma modification of these polymers can further improve their ability to deliver vaccines [123].

Plasma treatment can modify the surface properties of natural polymers, allowing for better interaction with antigens and adjuvants. Additionally, plasma treatment can increase the surface area of the polymer, leading to increased loading capacity of vaccine components [100]. For example, plasma-modified chitosan nanoparticles have been shown to enhance the immunogenicity of a hepatitis B vaccine by increasing the cellular uptake of the vaccine components [124]. Similarly, plasma-modified alginate microparticles have been shown to improve the stability of a DNA vaccine by protecting it from enzymatic degradation [125].

Another advantage of plasma-modified natural polymer-based drug delivery systems for vaccines is their ability to enhance mucosal immunity. The use of natural polymers as carriers can facilitate the transport of vaccines across mucosal surfaces, leading to a stronger immune response at the site of infection. For example, plasma-modified chitosan nanoparticles have been shown to enhance the immunogenicity of a nasal influenza vaccine [120].

Overall, plasma-modified natural polymer-based drug delivery systems have the potential to improve vaccine efficacy by enhancing immune responses, improving stability, and facilitating mucosal delivery. However, further research is needed to optimize plasma treatment conditions and evaluate the safety and efficacy of these systems in clinical settings (Table 2).

### 9.6. Plasma-Based 3D Printing for Fabrication of Complex Drug Delivery Systems

Plasma-based 3D printing is a type of additive manufacturing technology that uses plasma to create three-dimensional structures. In drug delivery systems, plasma-based 3D printing can be used to create drug delivery devices with precise control over the size, shape, and composition of the device [140]. The plasma-based 3D printing process involves the use of a plasma torch to melt and fuse together layers of material to create the final structure [141]. This technology has the potential to create drug delivery devices with complex geometries and controlled release properties, which can improve the efficacy and safety of drug delivery systems. However, more research is needed to fully understand the potential of plasma-based 3D printing in drug delivery systems [123]. In a study, researchers used plasma-based 3D printing to create microneedles for transdermal drug delivery [142]. The microneedles were made from a biodegradable polymer and were designed to release a drug over a period of several hours. The researchers found that the microneedles were effective at delivering a model drug in vitro and in vivo. Table 2 provides a comprehensive overview of the various applications and plasma modification techniques employed for natural polymers, offering valuable insights into their potential use in diverse fields.

## 10. Challenges and Future Directions

### 10.1. Poor Bioadhesion

Unmodified polymers often lack the necessary adhesive properties to adhere to biological tissues and surfaces. This can result in poor retention and reduced efficacy of drug delivery systems and biomedical devices [143]. Poor bioadhesion is a common challenge associated with unmodified polymers used in drug delivery systems [144]. Unmodified PLGA nanoparticles have been found to have poor mucoadhesive properties, which can limit their effectiveness in mucosal drug delivery applications [145]. Plasma modification techniques offer a valuable approach for altering the surface characteristics of polymers, thereby augmenting their bioadhesive attributes. This modification holds the potential to enhance the retention and effectiveness of drug delivery systems [38]. Plasma treatment has the capability to introduce functional groups, specifically amine, carboxyl, and hydroxyl groups, onto polymer surfaces, thereby augmenting their surface energy and enhancing their affinity towards biological surfaces [110].

### 10.2. Inadequate Biocompatibility

Some polymers may elicit an immune response or cause tissue irritation when used in biomedical applications [123]. This is a common limitation associated with unmodified polymers used in biomedical applications [146]. Unmodified chitosan can induce an inflammatory response and cause tissue damage in some cases [118]. Plasma modification techniques can be used to improve the biocompatibility of polymers by introducing biologically active functional groups to their surface [66]. For instance, plasma treatment can introduce amine and carboxyl groups to the surface of polymers, which can enhance their interaction with biological systems and reduce the risk of adverse effects [147].

### 10.3. Limited Drug Loading and Release Capacity

Unmodified polymers may have limited capacity to load and release drugs efficiently [148]. Unmodified polyethylene glycol (PEG) has limited drug loading and release capacity due to its low surface energy and hydrophobic nature [149]. Plasma modification techniques can be used to modify the surface properties of polymers and enhance their drug loading and release capacity [110]. Plasma treatment offers a means to incorporate functional groups, namely amines and hydroxyl groups, onto polymer surfaces, thereby enhancing their surface energy and hydrophilicity. This modification can result in improved drug loading and release characteristics of the polymers [146].

### 10.4. Lack of Specificity

Unmodified polymers may lack the necessary specificity to target specific cells or tissues, which can limit their effectiveness in drug delivery systems [150]. Unmodified PLGA nanoparticles can have limited targeting specificity, which can reduce their efficacy in targeted drug delivery applications [151]. Plasma modification techniques can be used to modify the surface properties of polymers and enhance their targeting specificity [146]. Plasma treatment can introduce functional groups, such as antibodies, peptides, and ligands, to the surface of polymers, which can improve their targeting specificity by enabling selective interactions with specific cells or tissues [143].

### 10.5. Limited Mechanical Strength

Some unmodified polymers may have inadequate mechanical strength, which can limit their use in biomedical applications that require durability and stability [152]. Unmodified poly(lactic acid) (PLA) can have limited mechanical strength, which can limit its use in load-bearing applications such as orthopedic implants [153]. Plasma modification techniques can be used to improve the mechanical strength of polymers by introducing cross-linking or reinforcing agents to their surface [39,85]. Plasma treatment can introduce functional groups, such as silanes or carbon nanotubes, to the surface of polymers, which can enhance their mechanical strength by increasing the intermolecular interactions between the polymer chains and reinforcing their structure [154].

### 10.6. Plasma-Modified Natural Polymer-Based Drug Delivery Systems for Cancer Therapy

While plasma modification offers several advantages for natural polymer-based drug delivery systems, there are also challenges that need to be addressed. Reproducibility and standardization of plasma treatment conditions are crucial for obtaining consistent and predictable modifications of natural polymers [5]. The plasma treatment parameters, such as power, gas composition, pressure, and treatment time, should be carefully controlled to achieve the desired modification [155]. Additionally, the effect of these parameters on the properties of the natural polymer should be systematically studied and optimized. Control over the surface properties and modification depth of natural polymers are additional challenges in plasma modification. The depth of plasma modification depends on the energy and flux of the plasma, which can be difficult to control precisely [80]. Moreover, the modification of surface properties may not always translate to the bulk properties of the polymer, which can affect its performance as a drug delivery system [115]. The effect of plasma treatment on the structural and chemical properties of the natural polymer is another challenge that needs to be considered. Plasma treatment can cause changes in the surface chemistry, morphology, and mechanical properties of the natural polymer, which can affect its biological activity and biocompatibility [156]. Therefore, it is important to carefully evaluate the effect of plasma treatment on the properties of the natural polymer and its drug delivery performance.

Finally, scale-up of plasma treatment for industrial applications is a challenge that needs to be addressed. Plasma treatment is typically performed on a small scale in the laboratory, and scaling up the process can be challenging due to the complexity and cost of plasma equipment. Moreover, the reproducibility and standardization of the process need to be ensured for large-scale production [157].

### 10.7. Emerging Trends in Plasma Modification of Natural Polymer-Based Drug Delivery Systems

Plasma modification of natural polymer-based drug delivery systems is a rapidly evolving field with several emerging trends. Plasma treatment can modify the surface properties of natural polymer-based drug delivery systems, such as chitosan, to enhance their biocompatibility, stability, and drug release properties [158]. For example, plasma treatment of chitosan-based films was shown to increase the surface area, improve the wettability, and enhance the drug release rate [158]. Similarly, plasma treatment of gelatin-based films was found to increase the surface roughness and hydrophilicity, resulting in improved cell adhesion and proliferation [159].

Plasma treatment can be used to introduce specific functional groups onto the natural polymer backbone, enhancing the drug delivery properties of the system. For example, plasma treatment of chitosan was used to introduce carboxyl groups, which were then used to conjugate doxorubicin, resulting in improved drug delivery efficiency [160]. Plasma treatment of hyaluronic acid was also used to introduce amine groups, which were then used for conjugation with paclitaxel, resulting in enhanced drug efficacy. Plasma treatment can be used to synthesize and functionalize nanoparticles, which can then be incorporated into natural polymer-based drug delivery systems, resulting in improved drug loading, stability, and controlled release properties [161]. For example, plasma-treated silver nanoparticles were successfully incorporated into chitosan-based drug delivery systems, resulting in improved antibacterial activity. Similarly, plasma treatment of graphene oxide was used to improve its dispersion in chitosan-based films, resulting in improved mechanical properties and controlled drug release [162].

### 10.8. Future Prospects and Opportunities

Plasma modification is a promising technique for improving the drug delivery properties of pharmaceutical formulations. Plasma modification can be used to tailor the drug delivery properties of formulations to individual patients [163]. This personalized approach can improve drug efficacy and reduce side effects. Plasma treatment can modify the surface properties of nanoparticles or natural polymers, allowing for targeted drug delivery to specific cells or tissues [164]. Plasma modification can be used to control the rate and duration of drug release from pharmaceutical formulations. Plasma treatment can modify the surface properties of nanoparticles or natural polymers, altering their interactions with drugs and controlling the release kinetics [165]. Plasma modification can enhance the bioavailability of poorly soluble drugs by improving their solubility and dissolution rate. Plasma treatment can modify the surface properties of nanoparticles or natural polymers, improving their interactions with drugs and enhancing drug solubility [163]. Plasma modification can be used to develop drug delivery systems for combination therapy, delivering multiple drugs simultaneously to treat complex diseases. Plasma treatment can modify the surface properties of nanoparticles or natural polymers, allowing for the co-delivery of multiple drugs with different properties [166]. Plasma modification can be used to enhance the properties of biomaterials used in tissue engineering, such as cell adhesion and proliferation. This technique can also modify the surface properties of natural polymers, such as collagen or gelatin, improving their interactions with cells and enhancing tissue regeneration [167].

## 11. Conclusions

Surface functionalization of natural polymer-based drug delivery systems can be achieved through the utilization of controlled plasma treatment, presenting a viable technique. This method allows for the modification of surface properties of natural polymers such as chitosan, cellulose, and gelatin, enhancing their interaction with pharmaceutical compounds and improving drug delivery characteristics. The precise modification of a natural polymer-based drug delivery system can be achieved through the application of adjustable plasma therapy, resulting in the creation of unique drug delivery systems that exhibit enhanced bioavailability, efficacy, and tailored administration. Moreover, plasma therapy demonstrates flexibility, scalability, and seamless integration into existing production processes for drug delivery systems. To advance the field of drug delivery and pharmaceutical development, researchers should prioritize the development of plasma sources and technologies specifically tailored for drug delivery, optimization of plasma treatment parameters, and exploration of natural polymer-based drug delivery approaches. Ultimately, the implementation of controlled plasma treatments holds significant potential for revolutionizing the landscape of drug delivery and facilitating pharmaceutical advancements.

## Figures and Tables

**Figure 1 pharmaceutics-15-02066-f001:**
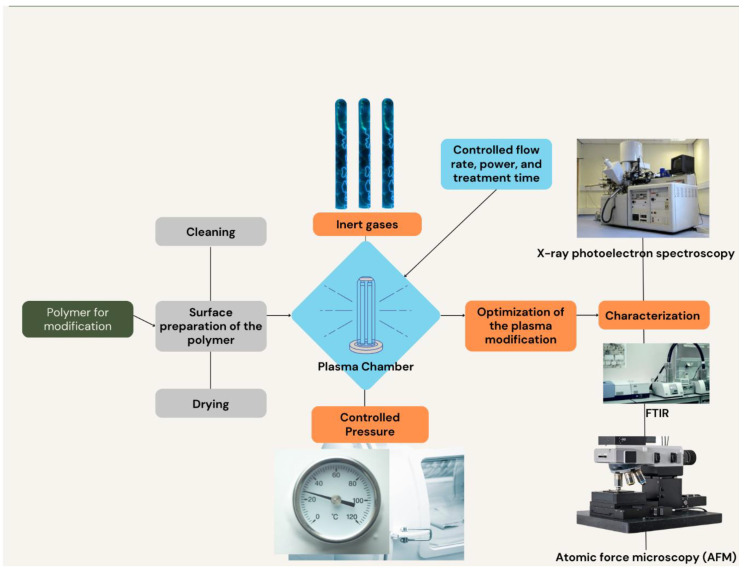
Schematic diagram of process of controlled plasma treatment.

**Figure 2 pharmaceutics-15-02066-f002:**
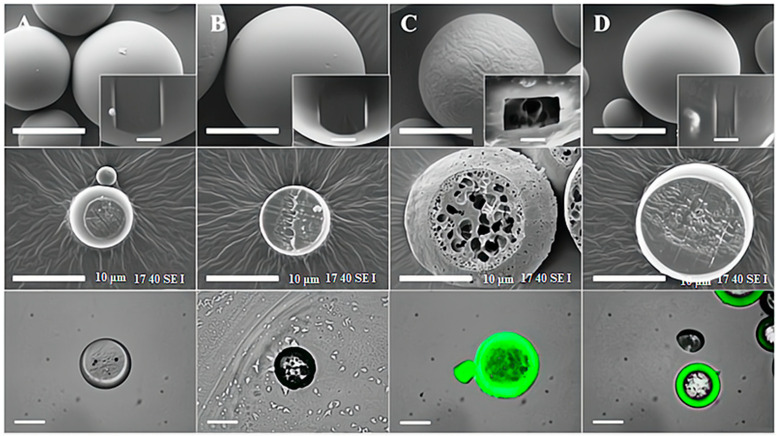
The SEM images presented in this study depict the PDLLGA microspheres prior to (**A**) and following modification with gel-MA using different methods, including surface adsorption (**B**), surface entrapment (**C**), and oxygen plasma treatment (**D**). Scanning electron microscopy (SEM) images reveal that the surface entrapment approach results in a porous structure, while the other approaches do not. Cross-sectioned SEM images demonstrate that the porous structure becomes increasingly smaller towards the surface in entrapment modified microspheres. Fluorescent images also indicate that entrapment modification leads to a deeper penetration of gel-MA compared to other approaches. Time-of-flight secondary ion mass spectrometry (ToF SIMS) ion peak spectra confirm the presence of gel-MA on all modified microspheres, with increasing ion counts of peptide bond related peaks. However, ion mapping images show that the distribution of gel-MA ions is homogeneous on plasma-modified microspheres, while it is heterogeneous on entrapment- and adsorption-modified microspheres. Non-modified PDLLGA microspheres do not exhibit any signs of gel-MA specific ions. All figures are presented with a scale bar of 50 μm, with magnified images at 5 μm [10].

**Figure 3 pharmaceutics-15-02066-f003:**
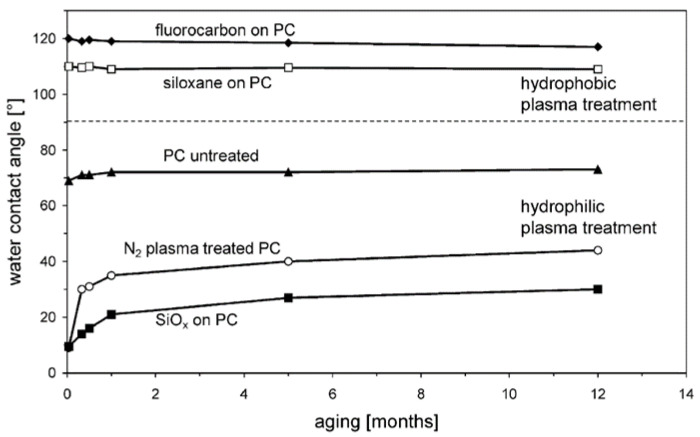
Figure illustrates the effect of plasma treatment on the wettability of aging untreated, plasma-treated, and coated PC sheets. The results show that the polycarbonate surface can be transformed from hydrophilic to hydrophobic, or vice versa, based on the plasma treatment.

**Table 1 pharmaceutics-15-02066-t001:** Enhancing Biocompatibility in Natural Polymer-Based Drug Delivery Systems.

Natural Polymer-Based Drug Delivery System	Plasma Treatment	Improvement in Biocompatibility	Reference
Chitosan Nanoparticles	Oxygen plasma	Reduced cytotoxicity and enhanced cellular uptake of drugs	[62]
Alginate Hydrogels	Argon plasma	Improved cell adhesion and proliferation	[63]
Starch-Based Films	Nitrogen plasma	Improved hydrophilicity, reduced contact angle, and increased cell adhesion	[64]
Cellulose Nanocrystals	Oxygen plasma	Improved cell proliferation and viability	[65]
Gelatin Microspheres	Argon plasma	Improved drug loading and release	[66]
Silk Fibroin Films	Oxygen plasma	Improved cell adhesion and proliferation	[67]
Hyaluronic Acid Nanoparticles	Oxygen plasma	Reduced cytotoxicity and enhanced biocompatibility	[68]
Pectin Hydrogels	Argon plasma	Improved cell adhesion and proliferation	[69]
Low-Density Polyethylene (LLDPE) and Poly(ethylene terephthalate) (PET) Films	Plasma source ion implantation (PSII) technique	Improved surface hydrophobic properties and increased contact angle and decreased surface energy observed	[70]
Ultra-High-Modulus Polyethylene Monofilaments	Oxygen plasma treatment	Improved adhesion to epoxy resins	[71]
Multi-Nanolayer Anticancer Drug	Low-pressure inductively coupled plasma (ICP)	Carboplatin and various other medications remained identifiable on the membrane for a duration exceeding 14 days in an artificial environment, while in a living organism, their detectability extended beyond a period of 10 days. The presence of a cytotoxic mesh resulted in a significant reduction in cellular adherence by a factor of 5.42 and provoked a substantial increase in the destruction of cancer cells by up to 7.87 times.	[72]
Plasma Polymerized Nanoparticles (PPNs)	Reactive gas discharges plasma treatment	PPNs carrying a combination of siVEGF and paclitaxel at reduced doses significantly reduced tumor growth in mice.	[73]
Tartary Buckwheat Starch	High-voltage and short-time (HV-ST) dielectric barrier discharge (DBD) plasma treatment	Improved solubility, paste clarity, in vitro digestibility, and decreased amylose content and viscosity.	[74]
PC, PP, EPDM, PE, PS, PET, and PMMA polymers	Ar, He, or N_2_ plasma treatments	Improved wetting, friction properties, and adhesion property.	[75]

**Table 2 pharmaceutics-15-02066-t002:** Applications of Plasma Modification Techniques of Natural Polymers in novel drug delivery systems.

Application	Natural Polymer	Plasma Modification	Reference
Gene Delivery	Chitosan	Improved hydrophilicity and charge density	[126]
Gene Delivery	Alginate	Increased surface area and introduction of functional groups	[127]
Gene Delivery	Hyaluronic Acid	Increased stability in the presence of hyaluronidase	[128]
Vaccine Delivery	Chitosan	Increased cellular uptake and immunogenicity	[129]
Vaccine Delivery	Chitosan	Facilitation of mucosal delivery and enhanced immunogenicity	[130]
Cancer Therapy	Chitosan	Enhanced cellular uptake and cytotoxicity	[131]
Wound Healing	Alginate	Improved cell proliferation and angiogenesis	[132]
Tissue Engineering	Hyaluronic Acid	Increased bioactivity and mechanical properties	[133]
Gene Therapy	Chitosan	Enhanced transfection efficiency	[134]
Anti-inflammatory Therapy	Gelatin	Increased drug loading and release	[135]
Dental Materials	Chitosan	Improved antibacterial properties	[136]
Skin Regeneration	Silk Fibroin	Improved mechanical properties and biocompatibility	[137]
Cardiovascular Disease	Hyaluronic Acid	Improved biocompatibility	[138]
Neurodegenerative Disease	Chondroitin Sulphate	Enhanced cell viability	[139]

## Data Availability

In this particular analysis or investigation, no new data has been generated or obtained. The existing data sources have been thoroughly examined and analyzed to provide insights and draw conclusions. Due to various factors such as time constraints, resource limitations, or data unavailability, it was not possible to gather additional data for this study. Despite the absence of new data, the findings and interpretations presented in this report are based on the available information and aim to contribute to the existing body of knowledge in this field.

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
