# Peer review of "Plasma Modification Techniques for Natural Polymer-Based Drug Delivery Systems"

_pharmaceutics, 2023, doi:10.3390/pharmaceutics15082066_

Round 1
Reviewer 1 Report
This paper is signifies the need of the drug delivery platforms using natural polymer platforms, I have few comments and suggestions to this paper,
1. Table 1 may be updated using other natural polymers in drug delivery and treatment aspects
2. The authors may include the relevant figures from the previously published papers to envisage their claim.
3. Schematic diagrams would be helpful for the paper to better understand.
4. The review manuscript should be update with the pre-literature published images and findings for further support the contents.
The English language may be improved in certain places for better understanding.
Author Response
Reviewer 1:
- Table 1 may be updated using other natural polymers in drug delivery and treatment aspects
Response:
|
Low density polyethylene (LLDPE) and poly(ethylene terephthalate) (PET) Films |
Plasma source ion implantation (PSII) technique |
Improved surface hydrophobic properties and increased contact angle and decreased surface energy observed |
[84] |
|
Ultra-high-modulus polyethylene monofilaments |
Oxygen plasma treatment |
Improved adhesion to epoxy resins |
[85] |
|
Multi-nanolayer anticancer drug |
Low-pressure inductively coupled plasma (ICP) |
Carboplatin and other drugs were detectable on the membrane for more than 2 weeks in vitro and more than 10 days in vivo. Cytotoxic mesh decreased cell adherence (down 5.42-fold) and induced cancer cell destruction (up to 7.87-fold). |
[86] |
|
Plasma polymerized nanoparticles (PPNs) |
Reactive gas discharges plasma treatment |
PPNs carrying a combination of siVEGF and Paclitaxel at reduced doses significantly reduced tumor growth in mice. |
[87] |
|
Tartary buckwheat starch |
High-voltage and short-time (HV-ST) dielectric barrier discharge (DBD) plasma treatment |
Improved solubility, paste clarity, in vitro digestibility and decreased amylose content and viscosity. |
[88] |
|
PC, PP, EPDM, PE, PS, PET and PMMA polymers |
Ar, He, or N2 plasma treatments |
Improved wetting, friction properties, and adhesion property. |
[63] |
- The authors may include the relevant figures from the previously published papers to envisage their claim.
Response: Previous published figures for the comparison has been added.
- Schematic diagrams would be helpful for the paper to better understand.
Response: Schematic diagrams for better understanding of the process has been added.
- The review manuscript should be update with the pre-literature published images and findings for further support the contents.
Response: The pre-literature published images and findings has been added for better understanding.

Reviewer 2 Report
The author has created a comprehensive review of the plasma treatment used for natural polymers. Fixing a few comments might help the manuscript
1) The limitations of current, unmodified polymers must be discussed a little more detail with references.
2) It is not clear in the introduction if the manuscript is discussing regarding plasma modifications in polymer or on the nanoparticle/hydrogel.
3) A graphical abstract summarizing the whole manuscript would help.
4) In the types of plasma source section. The author mentions about difference of plasma generated in different sources, it would help if they mention briefly the reason for the difference.
5) Both in sections 7 and 8 the author discusses the need/effect of plasma modifications, a brief description of caveats of current polymers needs to be discussed.
Author Response
1) The limitations of current, unmodified polymers must be discussed a little more detail with references.
Response: Poor bioadhesion: Unmodified polymers often lack the necessary adhesive properties to adhere to biological tissues and surfaces. This can result in poor retention and reduced efficacy of drug delivery systems and biomedical devices (Mogal et al., 2014). Poor bioadhesion is a common challenge associated with unmodified polymers used in drug delivery systems(Singh et al., 2020). Unmodified PLGA nanoparticles have been found to have poor mucoadhesive properties, which can limit their effectiveness in mucosal drug delivery applications (Surassmo et al., 2015). Plasma modification techniques can be used to modify the surface properties of polymers and enhance their bioadhesive properties, which can improve the retention and efficacy of drug delivery systems(Yoshida et al., 2013). Plasma treatment can introduce functional groups, such as amine, carboxyl, and hydroxyl groups, to the surface of polymers, which can increase their surface energy and improve their adhesion to biological surfaces(Petlin et al., 2017).
Inadequate biocompatibility: Some polymers may elicit an immune response or cause tissue irritation when used in biomedical applications(Sung and Kim, 2020). This is a common limitation associated with unmodified polymers used in biomedical applications(Pearce and O’Reilly, 2021). Unmodified chitosan can induce an inflammatory response and cause tissue damage in some cases (Nadesh et al., 2013). Plasma modification techniques can be used to improve the biocompatibility of polymers by introducing biologically active functional groups to their surface(Yoshida et al., 2013). For instance, plasma treatment can introduce amine and carboxyl groups to the surface of polymers, which can enhance their interaction with biological systems and reduce the risk of adverse effects(Gomathi et al., 2011).
Limited drug loading and release capacity: Unmodified polymers may have limited capacity to load and release drugs efficiently(Srivastava et al., 2016). Unmodified polyethylene glycol (PEG) has limited drug loading and release capacity due to its low surface energy and hydrophobic nature (Wang et al., 2023). Plasma modification techniques can be used to modify the surface properties of polymers and enhance their drug loading and release capacity(Petlin et al., 2017). Plasma treatment can introduce functional groups, such as amine and hydroxyl groups, to the surface of polymers, which can increase their surface energy and hydrophilicity and improve drug loading and release properties(Pearce and O’Reilly, 2021).
Lack of specificity: Unmodified polymers may lack the necessary specificity to target specific cells or tissues, which can limit their effectiveness in drug delivery systems(Sung and Kim, 2020). Unmodified PLGA nanoparticles can have limited targeting specificity, which can reduce their efficacy in targeted drug delivery applications (Li and Sabliov, 2013). Plasma modification techniques can be used to modify the surface properties of polymers and enhance their targeting specificity(Pearce and O’Reilly, 2021). Plasma treatment can introduce functional groups, such as antibodies, peptides, and ligands, to the surface of polymers, which can improve their targeting specificity by enabling selective interactions with specific cells or tissues(Petlin et al., 2017; Sung and Kim, 2020).
Limited mechanical strength: Some unmodified polymers may have inadequate mechanical strength, which can limit their use in biomedical applications that require durability and stability(Bolívar-Monsalve et al., 2021). Unmodified poly(lactic acid) (PLA) can have limited mechanical strength, which can limit its use in load-bearing applications such as orthopedic implants(Bikiaris et al., 2023). Plasma modification techniques can be used to improve the mechanical strength of polymers by introducing cross-linking or reinforcing agents to their surface(Petlin et al., 2017; Yoshida et al., 2013). Plasma treatment can introduce functional groups, such as silanes or carbon nanotubes, to the surface of polymers, which can enhance their mechanical strength by increasing the intermolecular interactions between the polymer chains and reinforcing their structure(Grace and Gerenser, 2003).
2) It is not clear in the introduction if the manuscript is discussing regarding plasma modifications in polymer or on the nanoparticle/hydrogel.
Response: The use of these natural polymers is limited by their poor solubility and stability in physiological conditions. To overcome these limitations, plasma modification techniques have been explored as a means of enhancing natural polymer-based drug delivery systems.
3) A graphical abstract summarizing the whole manuscript would help.
Response: Added
4) In the types of plasma source section. The author mentions about difference of plasma generated in different sources, it would help if they mention briefly the reason for the difference.
Response: The difference in plasma types and sources is mainly due to the differences in the physical and chemical properties of the plasmas generated and the materials being modified. Low-pressure plasmas are used for surface modification of materials because they operate at a lower pressure than atmospheric pressure, typically in the range of 0.1-10 Pa. At these pressures, the plasma can penetrate deeper into the material without causing significant damage to the surface. In addition, the plasma particles have a longer mean free path, which allows them to travel farther before colliding with a gas molecule or a surface. This means that the plasma can interact with the surface at a deeper level, leading to more effective surface modification. Atmospheric-pressure plasmas are preferred for large-scale surface modification because they can operate at or near room temperature and do not require a vacuum system. This is possible because atmospheric-pressure plasmas are generated using gas mixtures that can sustain a plasma discharge at atmospheric pressure without the need for a vacuum chamber. This allows for the treatment of large surface areas without the need for expensive vacuum equipment, which can be a limiting factor for industrial-scale surface modification processes. Additionally, because atmospheric-pressure plasmas operate at or near room temperature, they can be used to modify heat-sensitive materials without causing damage or altering their properties.
DC plasmas are simple to generate because they only require a DC power supply and a gas to ionize. The power supply can be as simple as a battery or a DC power supply unit, making them easy and inexpensive to set up. However, DC plasmas have limited control over their characteristics because the power supply and gas flow rate cannot be easily adjusted during operation. This can result in non-uniform plasma characteristics and difficulty in controlling the chemical reactions that occur at the plasma-material interface.
AC plasmas, on the other hand, are more complex because they require a power supply that can switch polarity at high frequencies. This is typically accomplished using a transformer and an AC power supply unit. While this adds complexity to the setup, AC plasmas offer greater flexibility and control over the plasma characteristics. The frequency and voltage of the power supply can be adjusted during operation, allowing for precise control over the plasma density and temperature. This results in a more uniform plasma and better control over the chemical reactions occurring at the plasma-material interface. RF plasmas are frequently utilized due to their capacity to create plasma with high density and uniformity, which is a critical factor for numerous applications such as plasma processing of semiconductors and surface modification of medical devices.
Capacitively coupled plasma (CCP) is a commonly used plasma source for surface modification due to its ability to generate a low-pressure plasma that can be easily controlled. In CCP, a high-frequency electric field is applied between two parallel plates, creating an electric discharge that ionizes the gas molecules and generates a plasma. The low pressure of the plasma and the ability to control the gas flow and power input make CCP ideal for surface modification of various materials. The resulting plasma can be used to modify the chemical and physical properties of a material surface, including its wettability, adhesion, and biocompatibility.
Recent advancements in plasma technology have led to the growing popularity of microwave plasma sources. These sources offer high energy efficiency and can generate plasmas at both low-pressure and high-density conditions. This makes them highly versatile and useful for a range of applications, such as materials processing, surface modification, and sterilization. Microwave plasma sources operate by generating an electromagnetic field at microwave frequencies, which excites the gas molecules and creates a plasma. The resulting plasma has a high degree of ionization and a uniform electron energy distribution, making it ideal for various industrial and scientific applications.
5) Both in sections 7 and 8 the author discusses the need/effect of plasma modifications, a brief description of caveats of current polymers needs to be discussed.
Response: Polymer materials have become ubiquitous in various fields due to their unique physical, chemical, and mechanical properties. However, many polymers have limitations that can hinder their performance in certain applications. For instance, poor adhesion to substrates, low surface energy, and inadequate biocompatibility are common issues encountered with many polymers. These shortcomings can limit their use in applications such as drug delivery, tissue engineering, and medical implants.
To overcome these limitations, various approaches have been proposed to modify the surface properties of polymers. Plasma modification is one of the most versatile and effective surface modification techniques that can improve the adhesion, wettability, and biocompatibility of polymer materials. By exposing the polymer surface to plasma, it is possible to introduce new functional groups, increase surface energy, and change the topography of the surface. These modifications can significantly enhance the performance of polymers in various applications.

Reviewer 3 Report
Nicely written review manuscript covering an interesting topic of surface modification of natural polymers for drug delivery.
I recommend it for publication in Pharmaceutics after minor revisions.
Authors should pay more attention to detail formating.
1. References within the text: Instead of (Yao et al., 2020), it would be appropriate to use [1]...
2. Reference list: The formating of references in the list is not according to the Pharmaceutics standard. Please, adjust accordingly.
3. Keywords formating: First keyword has both words capitalized, while the others have only the first word and teh second word starts in lower case. Surface Functionalization vs. Plasma modification, etc.
Figures and Tables are not mentioned in the text.
Author Response
- References within the text: Instead of (Yao et al., 2020), it would be appropriate to use [1]...
Response: The reference style has been changed.
- Reference list: The formatting of references in the list is not according to the Pharmaceutics standard. Please, adjust accordingly.
Response: The reference style has been changed.
- Keywords formatting: First keyword has both words capitalized, while the others have only the first word and then second word starts in lower case. Surface Functionalization vs. Plasma modification, etc.
Response: All keywords have been changed according to guidelines.
Figures and Tables are not mentioned in the text.
Response: All tables and figures are mentioned in the text.

Round 2
Reviewer 2 Report
Dear Author,
I see the comments have been addressed but not incorporated in the manuscript. Please make sure the correct version of the manuscript has been uploaded.
Author Response
Comments and responses of Reviewer-2
Comments and Suggestions for Authors
Dear Author,
Point 1. I see the comments have been addressed but not incorporated in the manuscript. Please make sure the correct version of the manuscript has been uploaded.
Response. The responses of all the following comments have been incorporated in revised manuscript and the changes have been made by track change. The changes in the revised manuscript have been shown by blue colour.
Reviewer 2:
1) The limitations of current, unmodified polymers must be discussed a little more detail with references.
Response: Poor bioadhesion: Unmodified polymers often lack the necessary adhesive properties to adhere to biological tissues and surfaces. This can result in poor retention and reduced efficacy of drug delivery systems and biomedical devices (Mogal et al., 2014). Poor bioadhesion is a common challenge associated with unmodified polymers used in drug delivery systems(Singh et al., 2020). Unmodified PLGA nanoparticles have been found to have poor mucoadhesive properties, which can limit their effectiveness in mucosal drug delivery applications (Surassmo et al., 2015). Plasma modification techniques can be used to modify the surface properties of polymers and enhance their bioadhesive properties, which can improve the retention and efficacy of drug delivery systems(Yoshida et al., 2013). Plasma treatment can introduce functional groups, such as amine, carboxyl, and hydroxyl groups, to the surface of polymers, which can increase their surface energy and improve their adhesion to biological surfaces(Petlin et al., 2017).
Inadequate biocompatibility: Some polymers may elicit an immune response or cause tissue irritation when used in biomedical applications(Sung and Kim, 2020). This is a common limitation associated with unmodified polymers used in biomedical applications(Pearce and O’Reilly, 2021). Unmodified chitosan can induce an inflammatory response and cause tissue damage in some cases (Nadesh et al., 2013). Plasma modification techniques can be used to improve the biocompatibility of polymers by introducing biologically active functional groups to their surface(Yoshida et al., 2013). For instance, plasma treatment can introduce amine and carboxyl groups to the surface of polymers, which can enhance their interaction with biological systems and reduce the risk of adverse effects(Gomathi et al., 2011).
Limited drug loading and release capacity: Unmodified polymers may have limited capacity to load and release drugs efficiently(Srivastava et al., 2016). Unmodified polyethylene glycol (PEG) has limited drug loading and release capacity due to its low surface energy and hydrophobic nature (Wang et al., 2023). Plasma modification techniques can be used to modify the surface properties of polymers and enhance their drug loading and release capacity(Petlin et al., 2017). Plasma treatment can introduce functional groups, such as amine and hydroxyl groups, to the surface of polymers, which can increase their surface energy and hydrophilicity and improve drug loading and release properties(Pearce and O’Reilly, 2021).
Lack of specificity: Unmodified polymers may lack the necessary specificity to target specific cells or tissues, which can limit their effectiveness in drug delivery systems(Sung and Kim, 2020). Unmodified PLGA nanoparticles can have limited targeting specificity, which can reduce their efficacy in targeted drug delivery applications (Li and Sabliov, 2013). Plasma modification techniques can be used to modify the surface properties of polymers and enhance their targeting specificity(Pearce and O’Reilly, 2021). Plasma treatment can introduce functional groups, such as antibodies, peptides, and ligands, to the surface of polymers, which can improve their targeting specificity by enabling selective interactions with specific cells or tissues(Petlin et al., 2017; Sung and Kim, 2020).
Limited mechanical strength: Some unmodified polymers may have inadequate mechanical strength, which can limit their use in biomedical applications that require durability and stability(Bolívar-Monsalve et al., 2021). Unmodified poly(lactic acid) (PLA) can have limited mechanical strength, which can limit its use in load-bearing applications such as orthopedic implants(Bikiaris et al., 2023). Plasma modification techniques can be used to improve the mechanical strength of polymers by introducing cross-linking or reinforcing agents to their surface (Petlin et al., 2017; Yoshida et al., 2013). Plasma treatment can introduce functional groups, such as silanes or carbon nanotubes, to the surface of polymers, which can enhance their mechanical strength by increasing the intermolecular interactions between the polymer chains and reinforcing their structure (Grace and Gerenser, 2003).
2) It is not clear in the introduction if the manuscript is discussing regarding plasma modifications in polymer or on the nanoparticle/hydrogel.
Response: The use of these natural polymers is limited by their poor solubility and stability in physiological conditions. To overcome these limitations, plasma modification techniques have been explored as a means of enhancing natural polymer-based drug delivery systems.
3) A graphical abstract summarizing the whole manuscript would help.
Response: Added
4) In the types of plasma source section. The author mentions about difference of plasma generated in different sources, it would help if they mention briefly the reason for the difference.
Response: The difference in plasma types and sources is mainly due to the differences in the physical and chemical properties of the plasmas generated and the materials being modified. Low-pressure plasmas are used for surface modification of materials because they operate at a lower pressure than atmospheric pressure, typically in the range of 0.1-10 Pa. At these pressures, the plasma can penetrate deeper into the material without causing significant damage to the surface. In addition, the plasma particles have a longer mean free path, which allows them to travel farther before colliding with a gas molecule or a surface. This means that the plasma can interact with the surface at a deeper level, leading to more effective surface modification. Atmospheric-pressure plasmas are preferred for large-scale surface modification because they can operate at or near room temperature and do not require a vacuum system. This is possible because atmospheric-pressure plasmas are generated using gas mixtures that can sustain a plasma discharge at atmospheric pressure without the need for a vacuum chamber. This allows for the treatment of large surface areas without the need for expensive vacuum equipment, which can be a limiting factor for industrial-scale surface modification processes. Additionally, because atmospheric-pressure plasmas operate at or near room temperature, they can be used to modify heat-sensitive materials without causing damage or altering their properties.
DC plasmas are simple to generate because they only require a DC power supply and a gas to ionize. The power supply can be as simple as a battery or a DC power supply unit, making them easy and inexpensive to set up. However, DC plasmas have limited control over their characteristics because the power supply and gas flow rate cannot be easily adjusted during operation. This can result in non-uniform plasma characteristics and difficulty in controlling the chemical reactions that occur at the plasma-material interface.
AC plasmas, on the other hand, are more complex because they require a power supply that can switch polarity at high frequencies. This is typically accomplished using a transformer and an AC power supply unit. While this adds complexity to the setup, AC plasmas offer greater flexibility and control over the plasma characteristics. The frequency and voltage of the power supply can be adjusted during operation, allowing for precise control over the plasma density and temperature. This results in a more uniform plasma and better control over the chemical reactions occurring at the plasma-material interface. RF plasmas are frequently utilized due to their capacity to create plasma with high density and uniformity, which is a critical factor for numerous applications such as plasma processing of semiconductors and surface modification of medical devices.
Capacitively coupled plasma (CCP) is a commonly used plasma source for surface modification due to its ability to generate a low-pressure plasma that can be easily controlled. In CCP, a high-frequency electric field is applied between two parallel plates, creating an electric discharge that ionizes the gas molecules and generates a plasma. The low pressure of the plasma and the ability to control the gas flow and power input make CCP ideal for surface modification of various materials. The resulting plasma can be used to modify the chemical and physical properties of a material surface, including its wettability, adhesion, and biocompatibility.
Recent advancements in plasma technology have led to the growing popularity of microwave plasma sources. These sources offer high energy efficiency and can generate plasmas at both low-pressure and high-density conditions. This makes them highly versatile and useful for a range of applications, such as materials processing, surface modification, and sterilization. Microwave plasma sources operate by generating an electromagnetic field at microwave frequencies, which excites the gas molecules and creates a plasma. The resulting plasma has a high degree of ionization and a uniform electron energy distribution, making it ideal for various industrial and scientific applications.
5) Both in sections 7 and 8 the author discusses the need/effect of plasma modifications, a brief description of caveats of current polymers needs to be discussed.
Response: Polymer materials have become ubiquitous in various fields due to their unique physical, chemical, and mechanical properties. However, many polymers have limitations that can hinder their performance in certain applications. For instance, poor adhesion to substrates, low surface energy, and inadequate biocompatibility are common issues encountered with many polymers. These shortcomings can limit their use in applications such as drug delivery, tissue engineering, and medical implants.
To overcome these limitations, various approaches have been proposed to modify the surface properties of polymers. Plasma modification is one of the most versatile and effective surface modification techniques that can improve the adhesion, wettability, and biocompatibility of polymer materials. By exposing the polymer surface to plasma, it is possible to introduce new functional groups, increase surface energy, and change the topography of the surface. These modifications can significantly enhance the performance of polymers in various applications.
